# Multiple risk factors are associated with an incremental increase in acute venous thromboembolism risk after total joint arthroplasty: A pearldiver cohort study

**Mandeep Kumar** [1,2] *, **Regina O. Kostyun**[2], **Matthew J. Solomito**[2], **Mitchell McClure**[1]

**1** Department of Hospital Medicine, Hartford Healthcare Medical Group, Hartford, Connecticut, United States of America, **2** Research Department, Hartford Healthcare Bone & Joint Institute, Hartford, Connecticut, United States of America

* Mandeep.kumar@hhchealth.org

## Abstract

### Introduction

Several risk factors are associated with acute venous thromboembolism (VTE) after total joint arthroplasty (TJA). However, there is a lack of literature regarding the cumulative impact of multiple risk factors. To address this gap, we utilized the PearlDiver database, an insurance billing claims database containing de-identified data from 91 million orthopedic patients.

### Methods

The PearlDiver database was queried for records of patients who underwent total hip and knee arthroplasty from 2010 to 2019 using ICD-10 and CPT codes. Twelve persistent and two transient risk factors were analyzed for their association with the occurrence of acute VTE within three months after surgery. Univariate and logistic regression analyses with odds ratios (ORs) and confidence intervals (CIs) were conducted to determine the odds associated with each risk factor and the impact of multiple concurrent risk factors.

### Results

A total of 988,675 patients who underwent hip and knee arthroplasty met the inclusion criteria, of whom 1.5% developed acute VTE after three months. The prevalence of VTE risk factors ranged from 0.2 to 38.6%. Individual, persistent risk factors demonstrated 14–84% increased odds of VTE compared to a 1.2% increase for a transient risk factor (acute myocardial infarction). Three or more persistent risk factors were associated with a higher risk of VTE.

### Conclusion and relevance

Persistent risk factors were associated with a higher incidence of postoperative VTE than transient risk factors. An incremental increase in risk was noted if three or more persistent risk factors were present.

**Data Availability Statement:** The data is owned by PearlDiver, a proprietary database containing de-identified and anonymized medical insurance billing

claims from 91 million orthopedic patients with Commercial, Medicare, and Medicaid insurance in the United States of America. Data cannot be shared publicly to protect patient privacy. Researchers may buy a subscription and analyze the data within their proprietary platform. https://pearldiverinc.com/contact/ Email - info@pearldiverinc.com.

**Funding:** This study was supported by an internal grant # 126736 from the Hartford Hospital medical staff (The funders had no role in study design, data collection and analysis, decision to publish, or preparation of the manuscript).

**Competing interests:** The authors have declared that no competing interests exist.

## Introduction

Total joint arthroplasty (TJA) promotes functional independence and improves the quality of life in patients with end-stage arthritis. Approximately 1.5 million lower extremity TJA procedures are performed in the United States annually [1]. A concerning postoperative complication is acute venous thromboembolism (VTE), estimated to occur in 0.6% to 3.0% of TJA cases [2]. Acute VTE events contribute to almost 100,000 deaths annually and add nearly $10 billion to healthcare costs [3]. Therefore, identification of at-risk patients is crucial to inform VTE prevention strategies.

The International Society on Thrombosis and Hemostasis suggests classifying VTE risks as persistent (i.e., age, weight, cancer diagnoses) or transient (e.g., surgical time, immobilization) [4]. Despite multiple studies evaluating VTE risk factors in patients with TJA, several gaps in literature remain. In a recent systematic review [5], older age [6–10], female sex [6], higher body mass index [9], bilateral procedures [9,11], and surgical times greater than two hours were reported as VTE risk factors. There are inconsistent findings or a lack of data on the role of diabetes mellitus [9,12], malignancy [9,12], VTE history [6,12], varicose veins [13], hypertension[9], and hormone replacement [14] on VTE risk. Some studies suggest that transient events, such as TJA [15,16] and acute myocardial infarction [17], may increase VTE risk for up to 6–12 months postoperatively. Several studies have examined these risk factors in isolation rather than the cumulative impact when more than one risk factor is present [8,18].

The primary goal of our study was to utilize the PearlDiver database to investigate the association between the cumulative effect of two or more VTE risk factors compared with TJA as an independent risk factor. National healthcare claims-based databases are valuable in providing the scale needed to explore low incidence events such as postoperative VTE, as this may not be feasible using data from one hospital or hospital system alone.

## Methods

A retrospective review was performed using PearlDiver, a proprietary database containing de-identified and anonymized medical insurance billing claims from 91 million orthopedic patients with Commercial, Medicare, and Medicaid insurance in the United States of America. This study was reviewed by the institutional review board of Hartford Healthcare and found to be exempt from review as the database utilized doesn't contain identifiable protected health information. Informed consent was not required due to the anonymized nature of the data.

Patients who underwent primary total hip or knee arthroplasty between 2010 and 2019 were identified using the Current Procedural Terminology (CPT) codes 27130 (arthroplasty, acetabular and proximal femoral prosthetic replacement, with or without autograft or allograft) and 27447 (total knee arthroplasty). Patients were included in the final analysis if they (1) were 40–85 years of age and (2) had continuous insurance coverage for a year before and after the index procedure. Data analysis was performed from 4/17/2023 to 4/21/2023.

The primary outcome was the occurrence of acute VTE within three months of the index procedure. Data queries used the International Classification of Disease 9th Revision (ICD-9) and 10th Revision (ICD-10) codes to identify patients who developed acute pulmonary embolism (ICD-9 41.511; ICD-10 I26.99) or acute lower extremity DVT (ICD-9 45.340; ICD-10 I82.4Z9, I82.4Z3, I82.4Z2, I82.4Z1, I82.4Y9, I82.4Y3, I82.4Y2, I82.4Y1, I82.499, I82.493, I82.492, I82.491, I82.409, I82.403, I82.402, I82.401).

Previously identified VTE risk factors from our literature review were investigated for their association with acute VTE after arthroplasty [5,19]. The ICD-9 and 10 codes used to identify the risk factors are outlined in S1 Table. Risk factors were categorized as transient (arthroscopic hip or knee surgery within the year leading up to arthroplasty surgery and acute

myocardial infarction) or persistent (70 years of age or older, sex, atrial fibrillation, cancer, congestive heart failure, hormone replacement therapy, hypertension, obesity, previous VTE, tobacco use, type 2 diabetes mellitus, and varicose veins).

Descriptive statistics were calculated for all demographic and outcome data. Binary and frequency-based data were presented as the percentage of the total study group to provide an epidemiological description of VTE after orthopedic surgery. Risk factors were grouped into one to eight concurrent risk factors. The incidence of acute VTE within three months after arthroplasty was compared for each risk factor using chi-square tests. Odds ratios (ORs) with 95% confidence intervals (CI) were calculated to assess the relative weights of each risk factor. Logistic regression analysis was performed to evaluate the association between cumulative number of risk factors and acute VTE. Statistical analyses were performed using R Studio software version 3.6.1 embedded within the PearlDiver platform. Statistical significance was set at $P < 0.05$.

## Results

A total of 988,675 patients who underwent total joint arthroplasty met the inclusion criteria: 14,827 patients developed acute VTE within three months of surgery, with an incidence of 1.5%. Of the patients with acute postoperative VTE, 72.8% (10,799 patients) underwent TKA and 27.2% (4,028 patients) underwent THA. Pulmonary embolism and deep vein thrombosis occurred in 20% (2,967) and 80.0% (11,860) of the patients, respectively.

The prevalence of the risk factors among all patients with TJA is shown in Table 1. Of the identified risk factors, nine were associated with increased odds of developing acute VTE after arthroplasty (Table 2). Risk factors for hip or knee arthroscopy within the year leading up to arthroplasty (100 patients, p = 0.203), history of hormone replacement therapy (28 patients, p = 0.433), female sex (9,119 patients, p = 0.213), and history of varicose veins (36 patients, p = 0.070) were not associated with the development of acute VTE. A history of VTE before arthroplasty was associated with lower odds of recurrent VTE (OR: 0.87, 95% CI: 0.79–0.96, p = 0.006), with 1.3% of patients with a history of VTE developing an acute VTE compared to 1.5% of patients without a history who went on to develop a VTE.

**Table 1. Prevalence of risk factors among all arthroplasty patients.**

| Risk Factor | | Arthroplasty Patients (%) | |
|---|---|---|---|
| *Persistent* | | Count | % |
| | Females | 603,172 | 61.0% |
| | Age $\geq$ 70 | 382,017 | 38.6% |
| | Hypertension | 145,856 | 14.8% |
| | Obesity | 68,185 | 6.9% |
| | Type 2 Diabetes | 52,003 | 5.3% |
| | Tobacco Use | 47,389 | 4.8% |
| | History of VTE | 32,686 | 3.3% |
| | Atrial Fibrillation | 21,953 | 2.2% |
| | Cancer | 19,712 | 2.0% |
| | Congestive Heart Failure | 14,134 | 1.4% |
| | Hormone Replacement Therapy | 2,162 | 0.2% |
| | Varicose Veins | 1,783 | 0.2% |
| *Transient* | | | |
| | Myocardial Infarction | 12,108 | 1.2% |
| | Arthroscopy | 5,880 | 0.6% |

**Table 2. Number of patients with persistent and transient risk factors and odds of developing Acute VTE after Hip and Knee Arthroplasty.**

| Risk Factor | | n | %[a] | OR | 95% CI | p |
|---|---|---|---|---|---|---|
| *Persistent* | | | | | | |
| | Congestive Heart Failure | 381 | 2.7% | 1.84 | 1.66–2.04 | <0.001 |
| | Obesity | 1284 | 1.9% | 1.29 | 1.21–1.36 | <0.001 |
| | Hypertension | 2,705 | 1.3% | 1.29 | 1.24–1.35 | <0.001 |
| | Atrial Fibrillation | 418 | 1.9% | 1.28 | 1.16–1.42 | <0.001 |
| | Type 2 Diabetes | 975 | 1.9% | 1.27 | 1.19–1.36 | <0.001 |
| | Cancer | 369 | 1.9% | 1.26 | 1.13–1.40 | <0.001 |
| | ≥ 70 years of age | 6371 | 1.7% | 1.20 | 1.16–1.24 | <0.001 |
| | Tobacco Use | 804 | 1.7% | 1.14 | 1.06–1.23 | <0.001 |
| | History VTE | 431 | 1.3% | 0.87 | 0.79–0.96 | 0.006 |
| *Transient* | | | | | | |
| | Acute MI | 218 | 1.5% | 1.21 | 1.05–1.38 | 0.006 |

[a]The percentage is based on the number of patients with the risk factor who developed an acute VTE after surgery.

A total of 7.0% (1,036 patients) of the patients who developed acute VTE had no additional risk factors (Table 3). Among patients with multiple persistent risk factors, with TJA without any risk factors as the baseline, three or more risk factors were associated with an increased risk of developing acute VTE after surgery, with a stronger association noted with every additional risk factor (Fig 1).

## Discussion

This study explored the association of multiple concurrent VTE risk factors with the occurrence of acute VTE after arthroplasty in a PearlDiver database sample of 988,675 adults in the US. We found that three or more concurrent risk factors were associated with a higher risk than two or fewer risk factors, and a progressive increase in risk was noted with each additional risk factor.

### Individual risk factors

Persistent risk factors were associated with higher odds of developing acute VTE. Patients with congestive heart failure had the highest odds of developing acute VTE within three months of

**Table 3. Impact of cumulative risk factors and odds of developing Acute VTE after Hip and Knee Arthroplasty.**

| Risk Factor count | | n | %[a] | OR | 95% CI | p |
|---|---|---|---|---|---|---|
| | TJA as independent risk factor | 1,036 | 0.5% | | | |
| | TJA + 1 | 1,141 | 1.4% | 0.61 | 0.58–0.64 | <0.001 |
| | TJA + 2 | 3,250 | 1.6% | 0.81 | 0.78–0.84 | <0.001 |
| | TJA + 3 | 3,970 | 1.8% | 1.04 | 1.01–1.08 | 0.01 |
| | TJA + 4 | 3,066 | 2.1% | 1.36 | 1.31–1.41 | <0.001 |
| | TJA + 5 | 1,569 | 2.4% | 1.53 | 1.45–1.62 | <0.001 |
| | TJA + 6 | 612 | 2.7% | 1.78 | 1.64–1.93 | <0.001 |
| | TJA + 7 | 154 | 3.0% | 1.92 | 1.63–2.26 | <0.001 |
| | TJA + 8 | 29 | 4.5% | 2.98 | 2.00–4.24 | <0.001 |

[a]The percentage is based on the number of patients with risk factors developing an acute VTE after surgery.

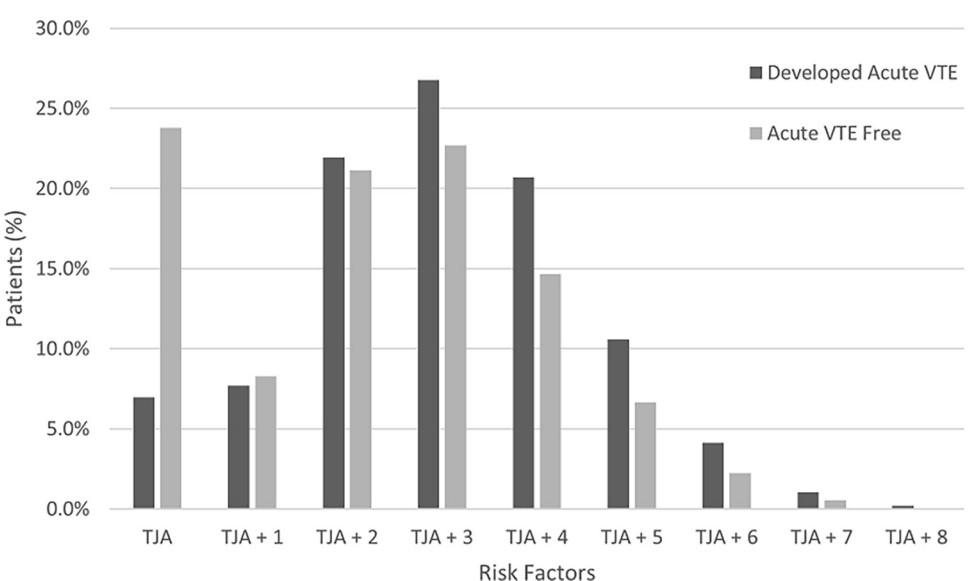

**Fig 1. Percentage of patients with one or more persistent risk factors, comparing those who developed acute venous thromboembolism (VTE) to those who remained VTE-free during the three months following total hip or knee arthroplasty (TJA).** Persistent risk factors in this analysis included age $\geq$ 70 years, cancer, congestive heart failure, obesity, persistent atrial fibrillation, tobacco use, type 2 diabetes mellitus, and hypertension.

surgery. Obesity, hypertension, persistent atrial fibrillation, type 2 diabetes mellitus, and cancer were individually associated with a 26–29% increase in the odds of developing acute VTE. Our results are supported by other studies reporting an association between older age [6,7,9,10], obesity [20], and atrial fibrillation with the incidence of acute VTE.

There is ambiguity regarding the roles of sex, diabetes mellitus, malignancy, varicose veins, hypertension, and hormone replacement as risk factors. Several studies have reported a variable association between sex and VTE risk factor among arthroplasty patients [10,17–19,21], whereas other authors found no association between sex and DVT [12,22]. We observed a similar incidence of acute VTE in males and females. Diabetes mellitus and hypertension are potential risk factors [19,23–25]. Diabetes mellitus and hypertension may not directly cause thrombosis but can be associated with cardiovascular disease, which may contribute to an increased risk.

Addressing potentially modifiable VTE risk factors may be a plausible strategy for reducing the burden of acute VTE; however, this was not the focus of the present study. We identified obesity, tobacco use, and type 2 diabetes mellitus as potential risk factors. Weight loss strategies [26], smoking cessation treatments [27], and diabetes management [28] may play a role in decreasing VTE risk, which should be studied in future research.

## Multiple risk factors

Arthroplasty is an independent transient risk factor for acute VTE. Anderson et al. [29] noted a higher risk of acute VTE with Hip and knee arthroplasty as an independent risk factor. Our results showed that TJA patients with no additional risk factors had similar odds of developing acute VTE compared to TJA patients with one or two risk factors; however, the risk increased when three or more persistent risk factors were present. Evaluating the cumulative impact of VTE risk factors on Acute VTE in a robust sample size was a unique aspect of our study.

Our study had several limitations secondary to the inherent limitations of utilizing a billing-based database. The Pearldiver database does not contain prescription data; therefore, we

were unable to assess whether chemical prophylaxis with enoxaparin or another anticoagulant was prescribed or its postoperative duration. The lower odds of acute VTE with established risk factors, such as hormone replacement therapy and prior history of VTE, might have been due to the use of chemical prophylaxis. Additional limitations include the lack of clinical context and details on the status of persistent risks. For example, a diagnosis of diabetes is available, but appropriate laboratory values such as hemoglobin A1c are not available; hence, it is not feasible to determine whether diabetes is well-controlled or poorly controlled. Finally, due to the retrospective design, there is a possibility of selection bias and residual confounding. Despite these limitations, we believe that the robust sample size provides strength and validity to our findings regarding the cumulative impact of multiple concurrent risk factors.

## Conclusion

This study adds to the current understanding of persistent and transient risk factors for patients undergoing lower-extremity total joint arthroplasty in a large sample size (988,675 patients). Most individual persistent risk factors increase the odds of acute VTE by 20–30%. Three or more concurrent persistent risk factors were associated with a higher increased risk of postoperative VTE, with an incremental increase in risk with each additional risk factor. To our knowledge, this is one of the first studies to report that the risk of VTE progressively increases in the presence of multiple risk factors.

## Supporting information

**S1 Table. International classification of disease versions 9 and 10 and current procedural terminology used to query risk factors.**
(DOCX)

## Author Contributions

**Conceptualization:** Mandeep Kumar, Mitchell McClure.

**Data curation:** Mandeep Kumar, Regina O. Kostyun, Matthew J. Solomito.

**Formal analysis:** Mandeep Kumar, Regina O. Kostyun, Matthew J. Solomito.

**Funding acquisition:** Mandeep Kumar.

**Investigation:** Mandeep Kumar, Matthew J. Solomito.

**Methodology:** Mandeep Kumar, Matthew J. Solomito, Mitchell McClure.

**Project administration:** Mandeep Kumar, Regina O. Kostyun, Matthew J. Solomito.

**Resources:** Mandeep Kumar, Regina O. Kostyun, Matthew J. Solomito, Mitchell McClure.

**Software:** Regina O. Kostyun, Matthew J. Solomito.

**Supervision:** Mandeep Kumar, Mitchell McClure.

**Validation:** Regina O. Kostyun, Matthew J. Solomito.

**Writing – original draft:** Mandeep Kumar, Regina O. Kostyun, Matthew J. Solomito, Mitchell McClure.

**Writing – review & editing:** Mandeep Kumar, Regina O. Kostyun, Matthew J. Solomito, Mitchell McClure.

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
