## [Decision Letter · Decision Letter 0]

8 Apr 2024

PONE-D-24-03155Multiple risk factors are associated with an incremental increase in Acute Venous Thromboembolism risk after Total Joint Arthroplasty: A PearlDiver Cohort StudyPLOS ONE

Dear Dr. Kumar,

Thank you for submitting your manuscript to PLOS ONE. After careful consideration, we feel that it has merit but does not fully meet PLOS ONE’s publication criteria as it currently stands. Therefore, we invite you to submit a revised version of the manuscript that addresses the points raised during the review process.

Please submit your revised manuscript by May 23 2024 11:59PM. If you will need more time than this to complete your revisions, please reply to this message or contact the journal office at plosone@plos.org. Please include the following items when submitting your revised manuscript:A rebuttal letter that responds to each point raised by the academic editor and reviewer(s). You should upload this letter as a separate file labeled 'Response to Reviewers'.A marked-up copy of your manuscript that highlights changes made to the original version. You should upload this as a separate file labeled 'Revised Manuscript with Track Changes'.An unmarked version of your revised paper without tracked changes. You should upload this as a separate file labeled 'Manuscript'.

We look forward to receiving your revised manuscript.

Kind regards,

Eyüp Serhat Çalık

Academic Editor

PLOS ONE

Journal Requirements:

Internal institutional funding from our hospital medical staff

4. In the online submission form, you indicated that Aggregate data available with a subscription to PearlDiver

Additional Editor Comments:

I congratulate the esteemed authors for this important study. Venous thromboembolism (VT) remains a common and important complication after total joint arthroplasty surgeries. I think it was important that your study revealed a significantly increased incidence of VT, especially when there are three or more cumulative risk factors. Although the insurance billing database you mentioned in the limitation section has limitations, we kindly request you to expand the results section by providing more detailed information about risk factors in line with the suggestions of the reviewers and discuss them in the discussion section. Good luck.

Reviewers' comments:

Reviewer's Responses to Questions

**Comments to the Author**

1. Is the manuscript technically sound, and do the data support the conclusions?

Reviewer #1: Partly

Reviewer #2: Yes

Reviewer #3: Yes

2. Has the statistical analysis been performed appropriately and rigorously? 

Reviewer #1: No

Reviewer #2: Yes

Reviewer #3: Yes

3. Have the authors made all data underlying the findings in their manuscript fully available?

Reviewer #1: Yes

Reviewer #2: Yes

Reviewer #3: Yes

4. Is the manuscript presented in an intelligible fashion and written in standard English?

Reviewer #1: Yes

Reviewer #2: Yes

Reviewer #3: Yes

5. Review Comments to the Author

**Reviewer #1**: Needs more elaboration on the risk factors , and almost the risk factors are not well categorized this format the paper would provides incomplete and even non specific information about DVT particularly and specifically about post operative DVT.

**Reviewer #2:** Acute venous thromboembolism(VTE) after total joint arthroplasty (TJA) was clinically very dangerous. Several risk factors are associated with it. t is very important to study the risk factors of the disease and carry out effective prevention, so this paper is of great significance and value to the clinic. The author's final conclusion was: persistent risk factors were associated with a higher incidence of postoperative VTE than transient risk factors. An incremental increase in risk was noted if three or more persistent risk factors were present. This is obvious and not particularly new. The article would be more meaningful if the various risk factors could be classified and the risk level or probability of each risk factor for VTE could be assessed.

**Reviewer #3:** This is an unprecedented and groundbreaking study that shows a risk score specific to total joint arthroplasty with respect to Acute VTE, which is good news for the field of total joint arthroplasty, where VTE is known to be common. The risk factors are expressed in terms of numbers, which is simplied and easy to understand. Independent factors have different risks, and I am not sure if this should be expressed in the same score. However, the simplified scores are extremely clear and easy to understand.

6. PLOS authors have the option to publish the peer review history of their article (what does this mean?). If published, this will include your full peer review and any attached files.

Reviewer #1: **Yes: **Aram Baram

Reviewer #2: No

Reviewer #3: No

---

## [Author Response · Author response to Decision Letter 0]

27 Jun 2024

Response to reviewers document attached

---

## [Editor Report · Decision Letter 1]

30 Jul 2024

Multiple risk factors are associated with an incremental increase in Acute Venous Thromboembolism risk after Total Joint Arthroplasty: A PearlDiver Cohort Study

PONE-D-24-03155R1

Dear Dr. Kumar,

We’re pleased to inform you that your manuscript has been judged scientifically suitable for publication and will be formally accepted for publication once it meets all outstanding technical requirements.

Kind regards,

Eyüp Serhat Çalık

Academic Editor

PLOS ONE
---

## [Editor Report · Acceptance letter]

1 Aug 2024

PONE-D-24-03155R1 

PLOS ONE

Dear Dr. Kumar, 

I'm pleased to inform you that your manuscript has been deemed suitable for publication in PLOS ONE. Congratulations! Your manuscript is now being handed over to our production team.

Kind regards, 

on behalf of

Dr. Eyüp Serhat Çalık 

Academic Editor

PLOS ONE